# Case Studies with the Contiki-NG Simulator to Design Strategies for Sensors’ Communication Optimization in an IoT-Fog Ecosystem

**DOI:** 10.3390/s23042300

**Published:** 2023-02-18

**Authors:** Antonio Marcos Almeida Ferreira, Leonildo José de Melo de Azevedo, Júlio Cezar Estrella, Alexandre Cláudio Botazzo Delbem

**Affiliations:** Department of Computer Systems, University of São Paulo, São Carlos 13566-590, SP, Brazil

**Keywords:** fog computing, hybrid sensor network, multi-criteria decision making

## Abstract

With the development of mobile communications and the Internet of Things (IoT), IoT devices have increased, allowing their application in numerous areas of Industry 4.0. Applications on IoT devices are time sensitive and require a low response time, making reducing latency in IoT networks an essential task. However, it needs to be emphasized that data production and consumption are interdependent, so when designing the implementation of a fog network, it is crucial to consider criteria other than latency. Defining the strategy to deploy these nodes based on different criteria and sub-criteria is a challenging optimization problem, as the amount of possibilities is immense. This work aims to simulate a hybrid network of sensors related to public transport in the city of São Carlos - SP using Contiki-NG to select the most suitable place to deploy an IoT sensor network. Performance tests were carried out on five analyzed scenarios, and we collected the transmitted data based on criteria corresponding to devices, applications, and network communication on which we applied Multiple Attribute Decision Making (MADM) algorithms to generate a multicriteria decision ranking. The results show that based on the TOPSIS and VIKOR decision-making algorithms, scenario four is the most viable among those analyzed. This approach makes it feasible to optimally select the best option among different possibilities.

## 1. Introduction

A Wireless Sensor Network (WSN) is composed of devices connected to the Internet of Things (IoT) with different constraints, such as memory, energy consumption, scalability, and network robustness. All these devices have specific communication roles and functions that define the network, also known as Low-Power and Lossy Network (LLN). They can be introduced in different layers of connectivity: cloud, fog, edge, or IoT devices [1].

Figure 1 shows responsive, ubiquitous, and mobile devices at the edge of the network (Edge Computing) that respond as events occur, from simple sensors and actuators to others provided with more robust computational capabilities. The connectivity between the IoT layer and the fog layer requires less computational power than the connectivity between the fog layer and the cloud.

The infrastructure, platform, and applications in fog are interrelated, and their respective computational characteristics are distinct between the layers. Together they represent a stacked architecture in which the data is pre-processed locally and then diffused to the adjacent upper layers.

Cloud computing is essential for IoT to be globally available and to increase its processing capacity. However, it is possible to use fog computing architecture to provide services while keeping latency low, reducing network load, and improving energy efficiency [3].

Fog computing has evolved as a promising solution that can bring cloud applications closer to IoT devices near the edge of the network, which is a characteristic that contributes to low latency and lower response time [4]. However, fog computing also introduces constraints in this service layer, such as ensuring that its services are efficiently available to different IoT devices since they have limitations and present new challenges regarding the computational and energy resources used.

This increase in the number of built IoT devices has boosted research about applications for areas such as traffic surveillance [5,6], environmental monitoring [7,8], smart cities [9,10,11], intelligent transport systems [12], and agriculture [13,14]. These applications require a reconfigurable architecture and environments that require different computing resources that can be used more efficiently at the edge of the network. Furthermore, according to the authors [15], “the location selected to install sensors significantly affects the amount of information extracted from the measured data”.

Several gateway architectures have been proposed over the years to manage multiple sensors. However, performance concerns are related to high communication latency or variations in traffic load demands on networks generated through device mobility. Therefore, some studies introduce IoT concepts with fog computing to deploy applications targeting placement, distribution, scalability, device density, or mobility support [16].

Urban mobility services have as their essence the use of IoT technologies. Some research focuses on proposing a model to select the correct subset of buses that maximizes the coverage of a city [17]. Others solve linear optimization problems related to vehicles that follow predetermined routes and, as a solution, propose strategies that use heuristics [18]. Finally, we can mention research that has the purpose of collecting data from sensors coupled to buses [19].

Multi-attribute decision-making methods are widely used to solve problems of fog node selection and fog gateway selection. Different Multiple Attribute Decision Making (MADM) algorithms, including SAW, TOPSIS, and VIKOR, are used to compose a rank among the existing alternatives.

In this work, we describe and analyze the application of the Simple Additive Weighting (SAW), Technique for the Order of Prioritization by Similarity to Ideal Solution (TOPSIS) and VIseKriterijumska Optimizacija I Kompromisno Resenje (VIKOR) algorithms in metrics related to network, application and IoT device. Thus, selecting the most viable place to deploy a set of IoT sensors belonging to an LLN in fog for the public transport service of São Carlos in São Paulo, Brazil, is possible.

The selection of IoT devices often does not consider characteristics related to infrastructure, implantation strategies, or optimization metrics. As a result, our contribution focuses on the following:Define the best location among the evaluated scenarios to install a set of IoT devices to a network based on MADM methods.Maximize the supported data load of the proposed fog network for the urban mobility scenario with low communication latency.

## 2. Related Work

Determining the most suitable location to install a set of IoT devices from a fog network based on multiple criteria is both important and challenging. Because this particularity directly impacts the efficiency of the fog network, making it possible to reduce costs associated with its implementation and maintenance [20]. Considering aspects related to processing power, energy consumption, and network communication are also essential. Due to this context, research has been aimed at optimizing a single-objective value [21], studies dealing with bi-objective values [22,23], and research dealing with problems that include deciding on multiple objectives.

However, single-objective optimization proposes to optimize only one objective, while several critical metrics can be underestimated. Therefore, we should consider multiobjective optimizations for the real world to be applied in environments involving NP-hard problems. In [24], a study is proposed for approaches based on services, resources, and fog applications to be applied in smart cities. The authors list the most relevant metrics based on a revised literary study.

Multicriteria decision-making algorithms (MCDM) solve problems involving a finite number of alternatives according to the characteristics of each method. In the IoT context, different MCDM techniques have been used. MADM approaches are applied in various application domains; for example, in the article [25], the authors propose a strategy that uses the Pareto Optimal technique to compare the selection quality of the SAW, TOPSIS, and VIKOR algorithms related to specific criteria of IoT devices.

They are also commonly used to select cloud services; for example, in the article [26], the authors apply MCDM methods to the problem of choosing geographic regions for the Amazon Web Service cloud. In addition, a comparative analysis of the obtained ranking is carried out and verified both the time complexity of the different MCDM methods applied and the robustness of the classification methods. In the article [27], the AHP method is used in conjunction with fuzzy logic to classify cloud services. A hybrid multi-attribute decision-making (MADM) model is assigned to decrease the execution time of the ranking of cloud services.

In the article [28], the authors propose an integrated MCDM approach based on TOPSIS and Best Worst Method (BWM) that uses evaluation criteria to classify the Cloud Service Provider according to the fulfillment of the customer’s requirements. The article [29] focuses on problems that evaluate and rank IoT applications using AHP and SAW algorithms. In the paper [30], the authors propose a more effective recommendation system to present IoT applications. Initially, they apply the AHP algorithm to evaluate and classify IoT applications. Then they assign a sequential quadratic programming algorithm to automatically find the optimal weight of the criteria and sub-criteria.

Other studies apply heterogeneous network selection mechanisms for the Internet of Vehicles (IoV) [31], and others expose a comparative study between fuzzy AHP and fuzzy TOPSIS techniques for the reliable and connected selection of cluster leaders in a mobile wireless sensor network [32]. In the article [33], a hybrid decision-making algorithm is implemented by merging the Fuzzy Analytic Hierarchy Process (FAHP) and Dynamic Analytic Hierarchy Process (DAHP) algorithms to be applied to Intelligent Transport Systems. Finally, we mention the article [34], which uses optimization methods for network selection based on various criteria covering quality of service, mobility, cost, energy, battery life, etc.

When analyzing Table 1, we observed that MADM methods are applied in different optimization problems over the available alternatives characterized by multiple, often conflicting, attributes. This list is not comprehensive but only representative. We mainly considered reviews or research articles in the context of our study.

## 3. Multiple Criteria Decision Making

Multi-criteria decision-making (MCDM) refers to choosing the best alternative among a finite set of decision alternatives that are affected by different, often conflicting, multiple criteria [35]. Based on the number of alternatives under consideration, the MCDM can be classified into:Multi-Attribute Decision Making (MADM): It is suitable for evaluating discrete decision spaces with predetermined decision alternatives. The MADM approach requires selecting a predetermined and limited number of decision alternatives. In addition to sorting and ranking, MADM approaches can be seen as alternative methods for combining information in a problem’s decision matrix with additional information from the decision maker to determine a final ranking or selection from among the alternatives [36].Multi-Objective Decision Making (MODM): It is preferably used for continuous decision problems where the alternatives are not predetermined. Instead of optimizing a goal function, it is focused on optimizing several goal functions.

An example of the classification of the MCDM is shown in Figure 2.

Multi-attribute decision-making algorithms are used in optimization problems that can be classified into scheduling, allocation, placement, offloading, load balancing, resource provisioning, selection, and others [37].

This article focuses on how to efficiently deploy devices in a fog network to efficiently service requests related to devices integrated into a public transport network based on multiple criteria and sub-criteria. The criteria may be dynamic or static and require maximization or minimization. For example, the latency criteria are related to network conditions and load. It is a dynamic criterion that must be minimized.

Many MADM techniques are presented in the literature, but the SAW, VIKOR, and TOPSIS methods are well-known and involve a simple computational process. The proposed methodology makes it possible to determine the location to deploy IoT sensors that best suit your circumstances and needs. However, it does not provide a universal and definitive solution. A brief description of each method is presented in the following subsections.

### 3.1. Simple Additive Weighting (SAW)

According to authors [38,39], the central concept of this method is to find the weighted sum of the performance evaluations of each alternative in all attributes, which requires the normalization process of the decision matrix (X) to a scale comparable to all alternatives to existing assessments.

This method is also referred to as the simplest and easiest to use among MADM methods. Mathematical formulation [40,41] is described to the following:The criteria used as a reference in the decision are specified and named in ($C_{i}$);It is necessary to determine the adjustment value of each alternative in each attribute;Make decisions based on the criteria in the array ($C_{i}$). The matrix is normalized according to the fitted equations for the attribute type (attribute or attribute benefit costs) to obtain the normalized matrix;The final result is obtained from the multiplication process of the classification matrix, which is the sum of the normalized R with the weight vector. This way, the highest value is obtained and selected as the best alternative ($A_{i}$) for the solution.

If *j* is an attribute benefit, we have Equation (1).
(1)$$r_{ij}=\left\{\frac{X_{ij}}{Max(X_{ij})}\right\}$$

If the attribute *j* is the cost, then use the formula (2).
(2)$$r_{ij}=\left\{\frac{Min(X_{ij})}{X_{ij}}\right\}$$

Observation:

$r_{ij}$ = Normalized value of the performance evaluation;

$X_{ij}$ = obtained value attribute. 

Criterion:

*Max*$X_{ij}$ = highest value obtained from each criterion;

*Min*$X_{ij}$ = lowest value obtained from each criterion;

Benefit = If the highest value is the best value;

Cost = If the lowest value is the best value. 

In the equation presented in (3), we have that, $r_{ij}$ is the value to be classified of the alternative $A_{i}$ in the attribute $C_{j}$; *i* = 1, 2 …, m and *j* = 1, 2 …, n. The value preferences for each alternative ($V_{i}$) are given as:(3)$$Vi=\sum_{j-1}^{n}W_{j}r_{ij}$$

Observation:

$V_{i}$ = Ranking of each alternative;

$W_{j}$ = Weight value of each criterion;

$r_{ij}$ = The ranked value $V_{i}$ shows that the highest value is the preferred alternative $A_{i}$.

### 3.2. Technique for the Order of Prioritisation by Similarity to Ideal Solution (TOPSIS)

The authors Hwang and Yoon [42] proposed the method of demand performance based on the correlation to the optimal solution (TOPSIS). It is a method that weighs several alternatives and criteria in a generalized situation. TOPSIS describes a solution with the shortest distance to the ideal solution, defined as Positive Ideal Solution (PIS), and the most significant distance from the negative ideal solution, defined as Negative-Ideal Solution (NIS). However, it does not consider the relative importance of these distances [43].

The TOPSIS algorithm can be successfully applied for decision-making in different study areas, including complex network analysis [44,45], Internet of Things [46,47,48], neural networks [49,50,51], reverse logistics [52,53], and sensor selection [54,55,56]. According to [57], the mathematical formulation of the TOPSIS algorithm is composed of the steps:The decision matrix *D* is represented as
$$D=\left[\begin{array}{cccc} X_{11} & X_{12} & & X_{1N}\\ X_{21} & X_{22} & & X_{2N}\\ X_{M1} & X_{M2} & & X_{MN} \end{array}\right]$$The elements $r_{ij}$ of the ordered decision matrix are calculated according to Equation (4).
(4)$$r_{ij}=\frac{x_{ij}}{\sqrt{\sum_{i=1}^{m}x{{}_{ij}}^{2}}}$$To generate the weighted ordered decision matrix, the corresponding weights $w_{n}$ of the different criteria are multiplied with the obtained values $r_{ij}$.
$$V=\left[\begin{array}{cccc} r_{11}W_{1} & r_{12}W_{2} & & r_{1N}W_{N}\\ r_{21}W_{1} & r_{22}W_{2} & & r_{2N}W_{N}\\ r_{M1}W_{1} & r_{M2}W_{2} & & r_{MN}W_{N} \end{array}\right]$$The *PIS* and the *NIS* are formulated according to Equations (5) and (6).
(5)$$PIS;\;A^{*}=\left\{\left(max\;v_{ij}|j\varepsilon J\right),\left(min\;v_{ij}|j\varepsilon j^{{}^{\prime }}\right)\right\}$$
(6)$$NIS;\;A^{-}=\left\{\left(min\;v_{ij}|j\varepsilon J\right),\left(max\;v_{ij}|j\varepsilon j^{{}^{\prime }}\right)\right\}$$
where *i* = 1, 2, 3 …. M e *j* = 1, 2, 3, …, NJ ∈ {Benefit Criteria Set}$J^{{}^{\prime }}$∈ {Cost Criteria}The distance of each alternative is calculated from the *PIS* and *NIS* according to Equations (7) and (8).
(7)$$P_{i^{*}}={\left(\sum {\left(v_{ij}-v_{j^{*}}\right)}^{2}\right)}^{1/2},i=1,2,3,4\ldots \ldots .M$$
(8)$$P_{i^{-}}={\left(\sum {\left(v_{ij}-v_{j^{-}}\right)}^{2}\right)}^{1/2},i=1,2,3,4\ldots \ldots .M$$The relative proximity of each alternative is calculated according to Equation (9).
(9)$$C_{i^{*}}=P_{i^{-}}/\left(P_{i^{*}}+P_{i^{-}}\right),0\leq C_{i^{*}}\leq 1,i=1,2,3,4,\ldots M$$Finally, the values of the proximity coefficient obtained with Equation (9) make it possible to calculate the ranking order.

### 3.3. VIseKriterijumska Optimizacija I Kompromisno Resenje (VIKOR)

According to [58], VIKOR “is a classification method for a finite set of alternative actions to be classified and selected among the criteria and solves a discrete multi-criteria problem with non-quantifiable and conflicting criteria”.

In the work of [59], the authors show that the VIKOR method is applied in several fields, such as construction administration, material selection, performance evaluation, health, supply chain, management of tourism, quality of service, sustainability, and others.

The multi-criteria evaluation to adjust the ranking was developed from $L_{p}$-metric (Equation (10)), and is used as an aggregation function in a programming adjustment method. The various alternatives of *k* (*k* = 1, …, n) are represented as $a_{1},a_{2},\ldots ,a_{n}$. For alternative $a_{k}$, the classification of criterion *j* is denoted by $f_{kj}$, that is, $f_{kj}$ is the value of *j* and criterion of the function for alternative $a_{k}$; *m* is the number of criteria (*j* = 1, 2, …, m).
(10)$$L_{p,k}={\left\{\begin{array}{c} \sum_{j=1}^{n}{\left[\begin{array}{c} w_{j}\left(\begin{array}{c} f_{j}^{*}-f_{kj} \end{array}\right)/\left(\begin{array}{c} f_{j}^{*}-f_{j}^{-} \end{array}\right) \end{array}\right]}^{p} \end{array}\right\}}^{1/p},1\leq p\leq \infty ;k=1,2,\ldots ,n.$$

Regarding the VIKOR method, $L_{1,k}$ and $L_{\infty ,k}$ are used to formulate sorting criteria. The solution obtained by $min_{k}S_{k}$ has a maximum group function (“majority” rule, shown with an average difference when *p* = 1), and the solution obtained by $min_{k}R_{k}$, with a minimum individual analysis of the “concurrent”.

The adjustment solution $F^{c}$ is a feasible solution closer to the ideal of $F^{*}$, and the term adjustment means an agreement established by mutual concessions, as illustrated in Figure 3. Where, $\Delta f^{1}=f_{1}^{*}-f_{1}^{c}$ and $\Delta f^{2}=f_{2}^{*}-f_{2}^{c}$.

The VIKOR algorithm has the following steps:Determines the best $f_{j}^{*}$ and worst $f_{j}^{-}$ values of all functions and criteria, *j* = 1, 2, …, m. If function *j* represents a benefit, then $f_{j}^{*}=max_{k}f_{kj}$ or adjust $f_{j}^{*}$ is the desired/desired level, $f_{j}^{-}=min_{k}f_{k}^{j}$ being the worst-level configuration $f_{j}^{-}$.Calculate the values $S_{k}$ and $R_{k}$, *k* = 1, 2, …, n, by the relations:$S_{k}=\sum_{j=1}^{m}w_{j}\;|f_{j}^{*}-f_{kj}\left|\;/\;\right|f_{j}^{*}-f_{j}^{-}|$, displayed as the average distance;$R_{k}=max_{j}\left\{\right|f_{j}^{*}-f_{kj}\left|\;/\;\right|f_{j}^{*}-f_{j}^{-}\left|\;j=1,2,\ldots ,m\right\}$, shows how the maximum distance to priority improves, where $w_{j}$ are the criteria weights.Calculates the value $Q_{j}$, *k* = 1, 2, …, n, by the relation$Q_{k}=v\left(S_{k}-S^{*}\right)/\left(S^{-}-S^{*}\right)+\left(1-v\right)\left(R_{k}-R^{*}\right)/\left(R^{-}-R^{*}\right)$, *k* = 1, 2, …, m (alternatives).where:$S^{*}=min_{k}S_{k}$ or leave $S^{*}=0$, desired level;$S^{-}=max_{k}S_{k}$ or leave $S^{-}=1$, worst level;$R^{*}=min\;R_{j}$ or leave $R^{*}=0$, desired level;$R^{-}=max\;R_{j}$ or leave $R^{-}=1$, worst level.Therefore, it is possible to rewrite $Q_{k}=vS_{k}+\left(1-v\right)R_{k}$, when $S^{*}=0$, $S^{-}=1$, $R^{*}=0$ and $R^{-}=1$. It is worth mentioning that *v* is introduced because it is the weight of the “majority of criteria” approach (or “the maximum utility of the group”), here *v* = 0.5.Rank the alternatives, sorted by the values *S*, *R*, and *Q*, in descending order. The result is three ordered lists.

## 4. Case Study

In this section, we present the method for selecting the most suitable place to install IoT devices for a public transport network, which is simulated using Contiki-NG considering three groups of main criteria. Three MADM methods rank the different scenarios proposed for installing the devices. Some relevant points that differentiate our work from those shown in Table 1 are

All data is collected at runtime during the simulation of the analyzed scenarios;All sensors are emulated, so it is possible to carry out simulations with different types of sensors and obtain results closer to the real world;The performance analysis of the fog network infrastructure is carried out before its implantation.MADM methods are applied to multiple criteria involving different layers of the conceptual communication architecture model.

### 4.1. Problem Presentation

There are open questions in research related to optimization problems in fog computing. Some studies address the issue of placing nodes in fog [60,61,62], and the literature explores the benefits of using MCDM methods [37,63]. It is important to emphasize that this type of procedure is not an easy task, as many architectures, protocols, devices, criteria, and approaches are involved in its selection.

We apply MADM methods to select the most suitable location for deploying IoT devices among 5 (five) possible scenarios presented for the city of São Carlos—SP. This choice is due to the existence of a main objective for the decision maker (DM), which is to reach the most favorable solution among a set of criteria. The deployment of IoT devices, both at the interstate bus terminal and the bus stops close to it, makes it possible to collect data from many buses with lower communication latency to receive data from sensors installed on the buses.

The selection of the most viable points for the installation of IoT devices also results in the reduction of future costs related to a new installation, configuration, and maintenance of the sensor network, in addition to directly impacting the total data load supported by the network in fog.

### 4.2. Experiment Execution

Different programs and tools were used to conduct extensive experiments and analyze the results. Said experiments were out using a virtual machine on the VMWare virtualization software, with a microprocessor that includes 6 CPU(s), 64 GB RAM, and a disk with a storage capacity of 200 GB. The software used contains the Ubuntu 18.04.6 LTS 64-bit operating system (Kernel 5.4.0-91-generic), Contiki-NG-release/v4.6-58-gaa6e26f43-dirty, MySQL Server 5.7, PHP 7.2.24, RStudio Build 461 and Minitab 19.2 (64-bit).

All sensors applied during the experiments were emulated in Cooja, network communication is simulated in Contiki-NG, and access to the sensors occurs through the HTTP protocol. The criteria influence the choice of the most suitable place for installing the IoT sensors and refer to the IoT (sensors), fog (network), and cloud (software) layers. The sub-criteria applied to the optimization problem are shown in Figure 4.

The maximum number of mobile sensors supported in the analyzed scenarios is 30. Above this value, there is a communication overhead. The scenarios presented in Figure 5 were divided into 2 (two) groups, one with 22 sensors and the other with 37 sensors. In both groups, seven static nodes are responsible for receiving and sending all data traffic from the fog network. Node 1 (Sink Node/Middleware) is also responsible for communication between fog and cloud networks.

Six (6) simulations were performed per scenario, with a time interval of 1 hour per simulation and a total of 30 hours of simulation for each group. The data collected via Hypertext Transfer Protocol (HTTP) communication at runtime during the simulation was performed using a script developed in PHP Hypertext Preprocessor, with data being inserted into a MySQL database. Then, the arithmetic mean of each sub-criterion was obtained to populate the data table to which the decision-making algorithms were applied.

All nodes were distributed within 100 m, all interconnected through a hierarchical architecture and allocated according to the geographic coordinates obtained through google maps. For each mobile node, the time of getting on and off was considered, in addition to the vehicle’s movement according to the direction of the traffic of the existing streets and other routes.

Another essential point relates to the configuration parameters used in Contiki-NG to simulate the already presented scenarios. The parameters used to run the tests are shown in Table 2.

In a scenario composed of a set of sensors, applying MADM algorithms to assist in decision-making regarding the location of these sensors is essential. The decision matrix (m × n) with the values of the m alternatives for the n criteria are present in Table 3, and the foundations of this approach are divided into three groups:Alternatives: A set of alternatives will be classified: the five different scenarios presented in Figure 5.Attribute set: Represents criteria used in the decision-making process. For each scenario, the sub-criteria are present in Figure 4.Weights: The weights for the sub-criteria used in the decision process are shown in Table 3.

The algorithms SAW, TOPSIS, and VIKOR were implemented in the R programming language and generated the results via RStudio software.

## 5. Results

The SAW method provides a simple approach to obtain the normalized and weighted decision matrix. Figure 6 presents scenario two as the best rated for the group of 22 nodes and scenario 1 for the group with 37 nodes.

In decision-making, the TOPSIS method is applied to order alternatives and select the scenario that denotes the best option among the five alternatives. The decision matrix present in Table 3 is normalized using Equation (4), and the final ranking result for the analyzed scenarios is present in Figure 7, with scenario four as the best option for a group of 22 nodes. The VIKOR method considers the alternative closest to the ideal solution. Therefore, the ranking in Figure 8 presents scenario four as the best option for a group of 22 nodes.

There are limitations regarding the number of requests supported when increasing the number of nodes to 37. Specific nodes have “bad” values, that is, very low values, which directly impacts applying the SAW method to these values. Said values are considered when ranking the results, making decision-making prone to error.

The results presented in Figure 7 and Figure 8 do not show the rank of these nodes because some have values that negatively influence the final result. This situation occurred because the data collected by these nodes suffered traffic overload, high packet loss, and increased latency in the communication between the sensors and the application layer over the HTTP protocol.

The results in Table 4 show that the SAW method tends to induce errors in decision-making, so it will not be considered. The most robust alternative after applying the TOPSIS and VIKOR methods for the group with 22 nodes because of the evaluated criteria and assigned weights is scenario 4.

## 6. Conclusions

The connectivity between the IoT layer and the fog layer has less computational power than the cloud, and a way to get better performance in a sensor network that encompasses IoT devices, wireless communication, and applications is through the use of algorithms of optimization. MCDM methods are successfully used in optimization problems the several areas. Because of this, we apply the SAW, VIKOR, and TOPSIS algorithms to a device positioning problem to define the most viable location for deploying an IoT sensor network.

After defining the normalized decision matrix and assigning weights to the different sub-criteria, the results show that scenario 4 is the best classified by the TOPSIS and VIKOR methods. Being the best-classified alternative by the TOPSIS method indicates that this scenario is the best in terms of classification index and for being the closest alternative to the ideal solution among the analyzed scenarios. In addition, being the best-ranked alternative by the VIKOR method indicates that it is closer to the ideal solution of the methods evaluated. Both methods have the same scenario selection reference for fog computing sensor network deployment.

It is essential to point out that MADM algorithms have relatively high complexity due to the multiple criteria considered. Therefore, it is essential to evaluate the criteria and sub-criteria more objectively. Selecting the best location using MADM techniques among the alternatives allows you to increase the accuracy of service communication and reduce costs related to future problems with the deployed infrastructure.

In the future, we propose expanding the research scope and applying MODM methods to solve device placement optimization problems on different types of sensors integrated into the network. Thus, it will be possible to deploy fog devices efficiently and offer services to massive IoT devices without violating end user Quality of Service (QoS) requirements.

## Figures and Tables

**Figure 1 sensors-23-02300-f001:**
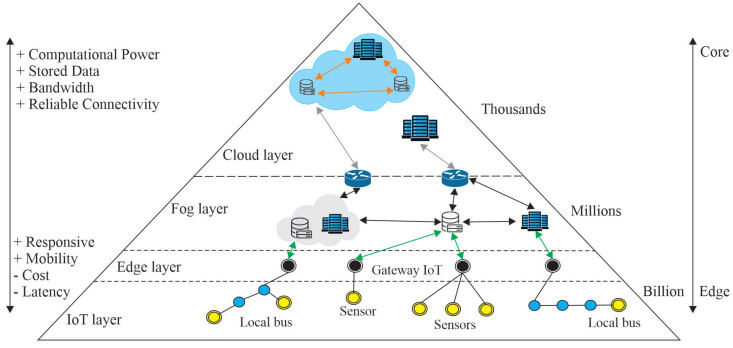
Conceptual Model of Communication Architecture. Adapted: [2].

**Figure 2 sensors-23-02300-f002:**
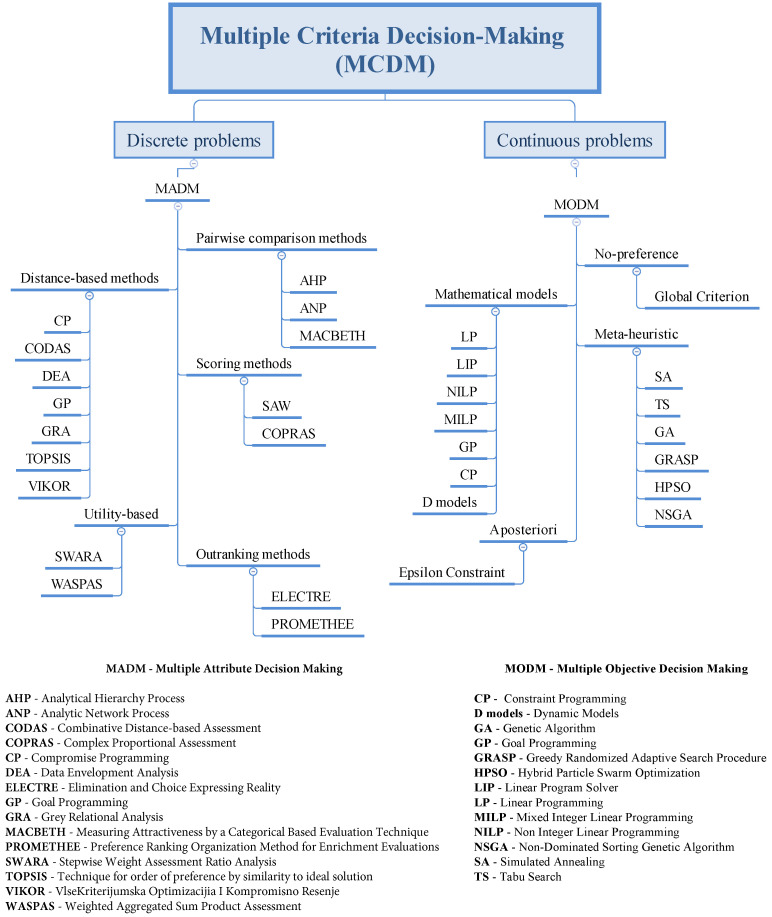
The classification of MCDM methods.

**Figure 3 sensors-23-02300-f003:**
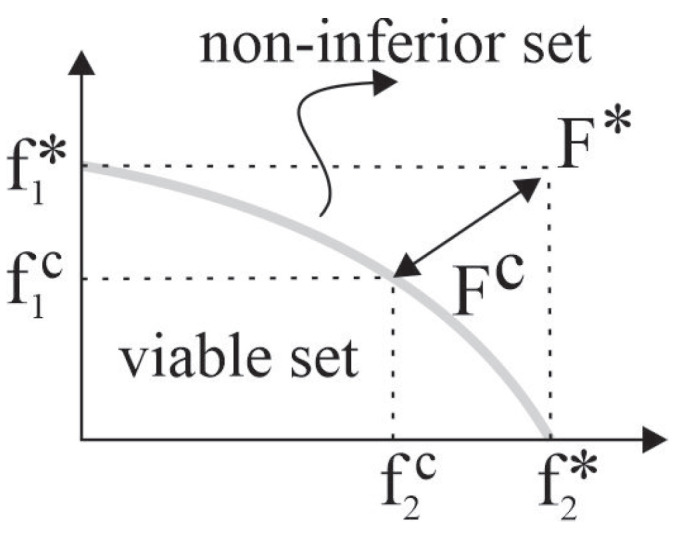
Optimal and Adjustment Solutions [58].

**Figure 4 sensors-23-02300-f004:**
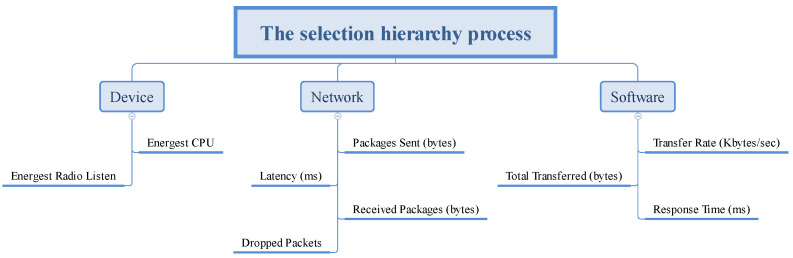
Metrics categorized into groups.

**Figure 5 sensors-23-02300-f005:**
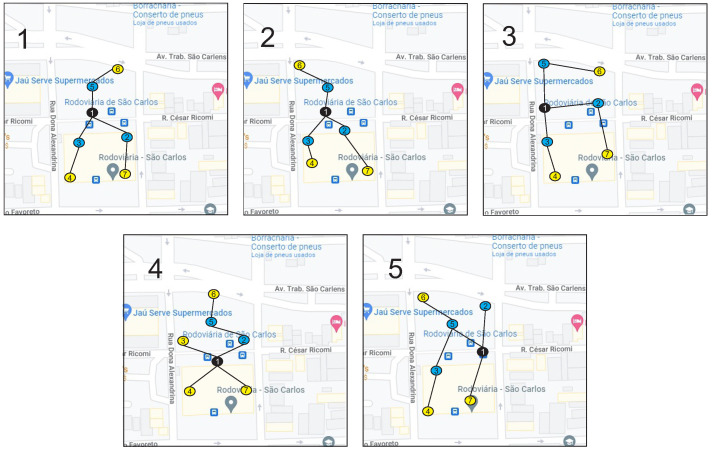
Experiment Scenarios on Google Maps.

**Figure 6 sensors-23-02300-f006:**
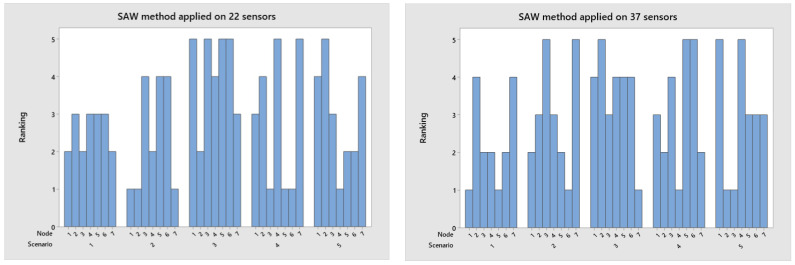
Rank applied SAW algorithm for the network with 22 and 37 nodes.

**Figure 7 sensors-23-02300-f007:**
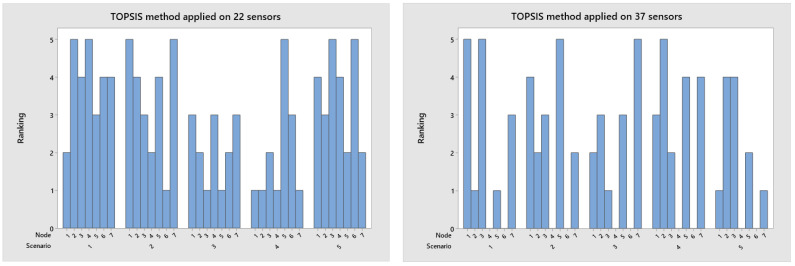
Rank applied TOPSIS algorithm for the network with 22 and 37 nodes.

**Figure 8 sensors-23-02300-f008:**
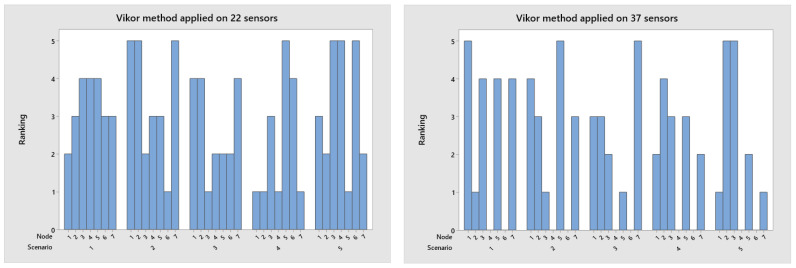
Rank applied VIKOR algorithm for the network with 22 and 37 nodes.

**Table 1 sensors-23-02300-t001:** Summary of studies taken under consideration.

Reference	Technique/Method	Algorithms	Main Criterion	Metric/Parameters of Evaluation	Application Areas	Year
[27]	Software based approach	AHP Fuzzy AHP	Accountability	-	Cloud Service	2022
				Capacity		
				Elasticity		
			Agility	Transparency		
				Availability		
				Interoperability		
				Service Stability		
				Serviceability		
			Assurance	Reliability		
			Cost	Service Cost		
				Service Response Time		
				Throughput		
			Performance	Accuracy		
			Security	-		
[26]	Software based approach	AHP PROMETHEE II TOPSIS VIKOR	Quality of Service (QoS)	Services	Cloud Service	2021
				Availability zone		
				Distance		
				Cost		
[30]	Software based approach	SQL Programming SAW ANP		Cost	IoT Applications	2021
				Energy Consumption		
			Smart Objects	Installation		
				Interoperability		
				Availability		
				Ease of Use		
			Application	Interface		
				Privacy		
				Reliability		
				Customer Care		
			Provider	Reputation		
				Number of Customers		
Proposed work	Hardware and Software based approach	SAW TOPSIS VIKOR		Energest CPU	Fog Service	2023
			Device	Energest radio listen		
				Packets sent		
				Packets received		
				Latency		
			Network	Lost packets		
				Response time		
				Transfer rate		
			Software	Total transferred		
[28]	Software based approach	AHP Hybrid (TOPSIS & Best-Worst Method)		Sustainability	Cloud Service	2020
				Interoperability		
			Performance	Service response time		
				Maintainability		
			Assurance	Reliability		
			Financial	Cost		
			Security & Privacy	Security Management		
			Agility	Scalability		
			Usuability	Usuability		
[25]	Software based approach	SAW VIKOR TOPSIS Pareto Optimal	Smart Objects	Battery	IoT Devices	2016
				Price		
				Drift		
				Frequency		
				Energy Consumption		
				Response Time		
[29]	Software based approach	AHP SAW		Cost	IoT Applications	2020
				Energy Consumption		
			Smart Objects	Installation		
				Interoperability		
				Availability		
				Ease of Use		
			Application	Interface		
				Privacy		
				Reliability		
				Customer Care		
			Provider	Reputation		
				Number of Customers		
[31]	Software based approach	AHP		Delay	Heterogenous Network	2021
				Packet loss rate		
			QoS	Bandwith		
				Jitter		
			Available load			
			Cost			
[32]	Software based approach	Fuzzy TOPSIS Fuzzy AHP	Cluster leader	Link Reliabililty	Cluster Leader Selection	2019
				Connectivity		
				Remaining Energy		
				Distance to BS		
				Speed		
[34]	Survey	SAW TOPSIS Weighted Product Model AHP GRA		Throughput	Network Selection	2019
				Delay		
			Application	Jitter		
				PLR		
				Energy consumption		
				Network load		
				Network coverage		
			Network	Network connection time		
				Available bandwidth		
				Battery level		
			Device	Mobility		
				Budget		
			User preferences	Cost		
[33]	Software based approach	Fuzzy AHP Dynamic AHP	Congestion control	Traffic flow	Intelligent Transportation Systems	2016
				Average speed		
				Occupancy rate		

**Table 2 sensors-23-02300-t002:** Parameter settings.

Parameters	Value
Simulation Tool	Contiki-NG
MAC	CSMA/CA
Transport	UDP/IPv6
Deployment type	Mobile and static position
Emulated nodes	Cooja
Simulation coverage area	1000 m × 1000 m
Total number of sensors	22–37
Fog Nodes	7
Sink Node	1
RX/TX ratio	100%
TX range	50 m
Interference range	100 m
Packet size	64 byte
Routing protocols	RPL Lite
Network protocol	IP based
Link failure model	UDGM with distance
Simulation time	60 min

**Table 3 sensors-23-02300-t003:** Decision Matrix.

Alternatives		Packets Send (bytes)	Latency (ms)	Packets Received (bytes)	Energest CPU	Packages Dropped	Energest Radio Listen (seconds)	Total Transferred (bytes)	Transfer Rate (Kbytes/sec)	Total Time (ms)
1	1	1151.6974	2294.9027	720.2119	1181.3946	0	74.9270	1823.4126	2.37	973.6
	2	375.7328	837.1118	473.1630	1187.8366	154	94.0973	237.5	0.11	4077.0727
	3	453.7193	1420.8156	552.4587	1187.1963	102	62.3146	238.0350	0.12	3936.5087
	4	215.5718	2003.5093	338.4059	1187.8366	345	145.7097	238.9423	0.03	14,211.7115
	5	432.9369	1515.3815	672.5302	1187.1963	6	56.2967	238.5535	0.13	4568.7321
	6	143.1311	1703.5139	218.7788	1187.8366	465	286.1891	239.4893	0.03	13,469.2340
	7	192.7308	2417.8158	296.8264	1187.8366	492	234.4047	239.6458	0.04	9813.0833
2	1	784.8094	2700.8251	487.9957	1082.0630	0	80.0212	1745.05454	1.78	1303.6
	2	226.2675	775.2763	264.6136	881.9036	18	53.9772	237.3	0.09	6020.2040
	3	510.7335	1357.9500	738.5428	882.33	106	79.3064	237.3695	0.11	4836.6739
	4	425.4716	2358.2422	698.2689	881.4670	76	118.6232	238.3953	0.02	17,349.5116
	5	253.7624	1049.4834	365.2913	881.1107	11	39.5167	237.6382	0.10	5232.2553
	6	154.1110	1578.6963	256.8010	879.3961	213	163.6175	238.8780	0.34	12,609.1463
	7	193.4476	2770.4797	318.7458	880.6006	256	152.5918	239.4102	0.03	19,391.6153
3	1	815.7785	2027.7524	500.8345	1246.3380	0	81.2630	1507.1166	2.55	658.2
	2	242.4766	1057.9882	289.9858	983.6701	0	51.5223	237.36	0.13	4646.4166
	3	347.3028	573.6886	400.6295	980.2740	0	36.6937	237.3333	0.15	2377.55
	4	117.8570	1957.6918	175.8373	953.0230	150	204.4628	239.4629	0.03	13,678.4444
	5	207.0744	667.3410	250.5757	1205.8702	0	37.2002	238.0344	0.14	2314.1896
	6	69.2964	1281.9875	104.0158	825.0438	139	182.1667	239.4905	0.03	10,657.5471
	7	140.7515	1718.0373	194.8677	790.0630	59	144.9888	239.2307	0.04	8596.9038
4	1	1465.2343	1410.2481	950.0392	1036.1140	0	90.3310	1810.45	2.52	717.5
	2	656.2387	764.3295	760.3923	1500.5576	6	92.3922	238.6800	0.14	2905.3684
	3	115.0719	650.2629	235.8162	1498.0567	0	62.8411	238.3448	0.15	2387.9827
	4	169.2818	740.2151	309.2612	1500.2273	2	87.0167	238.8596	0.11	4459.3684
	5	442.1057	1374.8193	534.3376	1545.5106	50	191.2584	239.8571	0.04	10,562
	6	175.0576	2292.8258	243.5753	1470.1182	298	368.7203	240.3962	0.02	20,272.9245
	7	167.1240	682.0284	298.2276	1520.906	2	93.6152	239.1250	0.15	2397.1964
5	1	785.4181	2270.7470	516.2583	1011.5442	0	89.6200	1832.4705	2.78	649.0
	2	63.7844	574.7394	112.5267	1052.1204	35	106.4600	238.25	0.15	3529.8958
	3	312.0038	2171.7712	381.2423	1040.6720	43	105.2626	238.7021	0.04	11,305.9787
	4	509.1450	3989.6861	932.4428	948.5080	83	168.8371	239.3709	0.02	23,748.3953
	5	550.9463	778.4576	619.0974	1302.0674	0	35.7709	238.0851	0.12	5945.7446
	6	67.2146	2202.9649	102.6245	1107.3020	583	314.5345	239.5	0.03	13,444.9166
	7	142.4695	741.2840	200.3583	920.2502	15	71.0000	239.0697	0.13	3168.8604
Weights		0.1	0.18	0.1	0.08	0.12	0.07	0.06	0.14	0.15

**Table 4 sensors-23-02300-t004:** Ranking results.

Alternatives		SAW	VIKOR	TOPSIS
1	1	1276.73	0.57079	0.460932
	2	989.37	1.00000	0.380069
	3	1200.24	0.70704	0.553937
	4	2679.43	0.00000	0.351706
	5	1171.98	0.67591	0.580214
	6	2529.96	0.37609	0.371579
	7	2098.98	1.00000	0.392042
2	1	1299.99	0.93492	0.048142
	2	1187.52	0.00000	0.499016
	3	802.67	0.08834	0.570139
	4	3200.01	0.25635	0.507123
	5	1121.16	0.18316	0.523249
	6	2319.02	0.03509	0.764470
	7	3528.51	0.20060	0.283881
3	1	1019.96	1.00000	0.406849
	2	1027.81	0.50000	0.544202
	3	699.24	0.23128	0.736800
	4	2524.19	0.08939	0.444361
	5	610.53	0.00000	0.619824
	6	1954.33	0.60163	0.437688
	7	1715.13	0.70929	0.580576
4	1	1124.54	0.34526	0.943643
	2	875.89	0.11585	0.826560
	3	1957.23	1.00000	0.665273
	4	1010.65	0.00000	0.673957
	5	2086.34	0.23784	0.361465
	6	3667.49	0.03955	0.375321
	7	691.96	0.22109	0.773653
5	1	1108.90	0.82969	0.379600
	2	761.31	0.87720	0.506544
	3	897.05	0.48848	0.326897
	4	4443.99	0.96344	0.435437
	5	1243.03	0.50000	0.611787
	6	2567.87	0.00000	0.279571
	7	743.32	0.19365	0.744092

## Data Availability

The article contains the data, which are also available from the corresponding author upon reasonable request.

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
