# Peer review of "Case Studies with the Contiki-NG Simulator to Design Strategies for Sensors’ Communication Optimization in an IoT-Fog Ecosystem"

_sensors, 2023, doi:10.3390/s23042300_

Round 1

Reviewer 1 Report

In this paper Authors simulate a hybrid network of sensors related to public transport in the city of São Carlos - SP using Contiki-NG. They collected the transmitted data based on the criteria corresponding to devices, applications, and network communication where they applied Multiple Attribute Decision Making (MADM) to generate a multi-criteria decision ranking. The topic is interesting and present well. However, I have following observations

1.      Author should mention their summary of results in abstract.

2.      Author should present a performance comparison with the algorithms SAW, TOPSIS and VIKOR in tabular form or in any kind of graph.

Reviewer 2 Report

This article reviews the basics of Wireless Sensor Networks (WSN) and how to integrate Edge Computing, Fog Computing and Cloud to minimize the latency for IoT device communication. Further, it analyses to apply the Multiple Attribute Decision Making (MADM) algorithms for selecting the most suitable places for implementing IoT nodes for the public transport service of Sao Carlos in Sao Paulo, Brazil. Although the article is interesting and provides a practical implementation but lacks several aspects.

1.      The related work section is very weak. It does not thoroughly review the existing works in the domain. More related works should add that fit the domain.

2.      The related works and their contributions should be highlighted in tabular form to easily summarize their contribution and how the proposed research work is different from theirs.

3.      The background about Multi-criteria decision-making (MCDM) is good, but the article should also discuss other possible options and the justification for choosing and applying MCDM should be elaborate.

4.      The references are not up to date and need to add more latest references.

5.      The role of different layers, like the fog layer, edge layer and cloud layer, has not been defined in the experimental setup and conclusion. It only discusses the IoT sensor deployment and optimization.

6.      The conclusion section should be more elaborate and include the future work section.

7.      Article should also highlight its novelty and contribution that is lacking except a simulated experiment given for a specific scenario but lacks in overall scientific contribution.

Round 2

Reviewer 2 Report

Authors have sufficiently improved their work based on the suggested review comments and can be accepted for publication in the current form.